# ER-associated mitochondrial division links the distribution of mitochondria and mitochondrial DNA in yeast

Andrew Murley[1], Laura L Lackner[1†], Christof Osman[2†], Matthew West[3], Gia K Voeltz[3], Peter Walter[2,4], Jodi Nunnari[1*]

[1]Department of Molecular and Cellular Biology, University of California, Davis, Davis, United States; [2]Department of Biochemistry and Biophysics, University of California, San Francisco, San Francisco, United States; [3]Department of Molecular, Cellular and Developmental Biology, University of Colorado, Boulder, Boulder, United States; [4]Howard Hughes Medical Institute, University of California, San Francisco, United States

**Abstract** Mitochondrial division is important for mitochondrial distribution and function. Recent data have demonstrated that ER–mitochondria contacts mark mitochondrial division sites, but the molecular basis and functions of these contacts are not understood. Here we show that in yeast, the ER–mitochondria tethering complex, ERMES, and the highly conserved Miro GTPase, Gem1, are spatially and functionally linked to ER-associated mitochondrial division. Gem1 acts as a negative regulator of ER–mitochondria contacts, an activity required for the spatial resolution and distribution of newly generated mitochondrial tips following division. Previous data have demonstrated that ERMES localizes with a subset of actively replicating mitochondrial nucleoids. We show that mitochondrial division is spatially linked to nucleoids and that a majority of these nucleoids segregate prior to division, resulting in their distribution into newly generated tips in the mitochondrial network. Thus, we postulate that ER-associated division serves to link the distribution of mitochondria and mitochondrial nucleoids in cells.

**\*For correspondence:**
jmnunnari@ucdavis.edu

†These authors contributed equally to this work

**Reviewing editor**: Richard J Youle, National Institute of Neurological Disorders and Stroke, United States

## Introduction

The distribution of mitochondria and mitochondrial DNA (mtDNA) is accomplished through the engagement of multiple pathways, including mitochondrial division, fusion, motility, and tethering. Mitochondrial division is mediated by a dynamin-related protein, Dnm1 (in yeast)/Drp1 (in mammals) (*Lackner and Nunnari, 2009*), which self-assembles in a regulated manner on the surface of the mitochondrial outer membrane into helices that mediate mitochondrial scission (*Ingerman et al., 2005*; *Mears et al., 2011*). Dnm1 and Drp1 form helices in vitro, whose diameters are significantly smaller than the diameter of unconstricted mitochondria (*Ingerman et al., 2005*; *Bossy et al., 2010*). This observation suggests that a mechanism for Dnm1/Drp1-independent constriction may exist to facilitate helix assembly as a first step toward mitochondrial division.

We recently discovered that ER tubules wrap around mitochondria and mark a majority of mitochondrial division sites in yeast and mammalian cells (*Friedman et al., 2011*). These findings suggest that the process of 'ER-associated Mitochondrial Division' (ERMD) facilitates the creation of mitochondrial constriction sites, or geometric 'hot spots', for Dnm1/Drp1 helix assembly. Consistent with this model, the association of ER tubules with mitochondrial constriction sites is independent of mitochondrial division components (*Friedman et al., 2011*). ERMD is also independent of known ER tubule-shaping proteins and Mfn2-mediated ER–mitochondria contacts (*de Brito and Scorrano, 2008*; *Friedman et al., 2011*). Indeed, the composition and biogenesis of ERMD sites are unknown.

**eLife digest** Mitochondria generate most of the energy used by cells, and they also play key roles in cellular growth, death, and differentiation. They are evolutionarily derived from bacteria and have retained their own DNA and protein translation system, but they are also dependent on the cell for their growth and replication.

A significant portion of the outer membrane of a mitochondrion is in contact with the endoplasmic reticulum (ER)—an organelle that is the starting point for the synthesis of secreted proteins, and is also critical for the synthesis of lipids and other organelles. Recent work suggests that mitochondria–ER contact points mark sites of mitochondrial division, but it is unclear exactly how this process occurs.

Here, Murley et al. use the budding yeast and model organism *Saccharomyces cerevisiae* to show that at mitochondrial division sites, a multiprotein complex called ERMES promotes the formation of ER–mitochondrial contact points, while an evolutionarily conserved enzyme, Gem1, antagonizes these contacts to aid mitochondrial segregation. The contact points are found adjacent to nucleoids (which are complexes of mitochondrial DNA and proteins)—an observation suggesting that ER-associated mitochondrial division evolved to help distribute nucleoids between newly formed mitochondria.

The present study also reveals a novel role for the conserved protein Gem1 and could lead researchers to reinvestigate the functions of Miro1/2—the equivalent of Gem1 in higher eukaryotes. Miro1/2 is thought to connect mitochondria to motor proteins, which transports them through the cell along microtubules. Dysfunction of Miro1/2 reduces the mobility of mitochondria, and the work of Murley et al. suggests that this could be a consequence of enhanced contacts between mitochondria and the ER.

A candidate for tethering ER and mitochondria at ERMD sites is the ER-Mitochondria Encounter Structure (ERMES)—a multiprotein complex localized at an interface between the ER and mitochondria in budding yeast cells (*Kornmann et al., 2009*; *Toulmay and Prinz, 2012*). The ERMES complex is composed of four core subunits, each of which is required for the formation of multiple ERMES foci per cell: Mdm10 and Mdm34 are integral to the mitochondrial outer membrane; Mdm12 is predicted to be cytosolic; and Mmm1 is an ER transmembrane protein (*Kornmann et al., 2009*; *Stroud et al., 2011*). ERMES is thought to function as a physical ER–mitochondria tether that serves to distribute mitochondria (*Boldogh et al., 2003*) and to facilitate the exchange of lipids between the two organelles (*Kornmann et al., 2009*; *Voss et al., 2012*). ERMES foci are localized adjacent to a subset of nucleoids, which are engaged in replicating DNA (*Hobbs et al., 2001*; *Hanekamp et al., 2002*; *Meeusen and Nunnari, 2003*), suggesting that ERMES plays an active role in nucleoid regulation. In agreement with this notion, deletion of core ERMES components disrupts nucleoid structure and transforms the tubular mitochondrial network into spherical mitochondria with relatively large diameters (*Burgess et al., 1994*; *Youngman et al., 2004*). The ERMES-deficient mitochondrial morphology phenotype is epistatic to the characteristic net-like mitochondrial structures that result from the deletion of genes encoding mitochondrial division components (*Youngman et al., 2004*), consistent with ERMES functioning upstream of the mitochondrial division machinery in mitochondrial distribution.

Cytological, biochemical, and genetic data indicate that the highly conserved Miro GTPase Gem1 is associated with ERMES at steady state (*Kornmann et al., 2011*; *Stroud et al., 2011*). In yeast cells lacking Gem1, mitochondria form a distinct spectrum of abnormal structures, from tubules to clustered spheres, mitochondrial distribution into daughter cells is less efficient than that of wild-type mitochondria, and cells lose mitochondrial DNA at a significantly higher frequency (*Frederick et al., 2004*; *Koshiba et al., 2011*). Gem1, however, is not an essential structural component of ERMES, as ERMES foci are observed in *gem1Δ* cells. However, in *gem1Δ* cells, it has been reported that ERMES foci are larger in size and fewer in number (*Kornmann et al., 2011*), suggesting that Gem1 acts as a regulator of ERMES.

Here, we show that ERMES and Gem1 are spatially and functionally linked to ERMD. Our data suggest a model where ERMES and Gem1 function in ERMD to facilitate the engagement and resolution

of ER–mitochondria contacts, respectively. Our data indicate that Gem1 acts as a negative regulator of ER–mitochondria contacts at ERMD sites and is required for the spatial resolution of newly formed mitochondrial tips generated by a division event. We also provide evidence that ERMD positions division sites adjacent to mitochondrial nucleoids to bias their distribution into newly generated tips following division. Thus, we postulate that ERMD evolved to link the distribution of nucleoids and mitochondria.

## Results

To assess whether ERMES foci are spatially linked to mitochondrial division, we simultaneously imaged fluorescent protein (FP)–tagged ERMES components and mitochondria over time in living yeast cells. To this end, we expressed functional C-terminal GFP fusions of Mdm12, Mdm34, or Mmm1 from their respective endogenous loci and labeled mitochondria using mitochondrial matrix-targeted DsRed (mito-DsRed). We observed a majority of mitochondrial division events associated with Mdm12, Mdm34, or Mmm1-labeled ERMES foci, which subsequent to the division segregate to only one of the two newly generated tips in the mitochondrial network (*Figure 1A,B*). The fraction of division events associated with ERMES foci (54–60%) is significantly higher than that predicted for a random association (approximately 10%), based on the surface area of mitochondria associated with ERMES (*Figure 1B* and 'Materials and methods'). The detection of ERMES foci by fluorescence imaging may underestimate ERMES foci associated with mitochondrial division as we observed a positive correlation between the percentage of mitochondrial divisions associated with a given ERMES subunit and its estimated whole-cell abundance (*Huh et al., 2003*). However, we cannot rule out the possibility that non-ERMES–associated mitochondrial division sites exist in cells.

In light of a previous report suggesting that the ERMES components Mdm34 and Mmm1 are not strictly colocalized (*Youngman et al., 2004*), we also addressed whether different types of ERMES complexes/foci exist that could be preferentially associated with mitochondrial division events. Examination of differentially FP-tagged Mdm34 and Mmm1 or Mdm12 indicated that all components colocalize at mitochondrial division sites (*Figure 1—figure supplement 1*), consistent with the finding that each core ERMES subunit is necessary for assembly of the complex (*Kornmann et al., 2009*). In addition, these two components associated with the same mitochondrial tip generated by division, indicating that the complex remains intact during the division process (*Figure 1—figure supplement 1*). Together, these observations demonstrate that mitochondrial division events are spatially linked to the ERMES complex.

We assessed the relationship of ERMES to ERMD by imaging cells expressing Mdm34-yEGFP, an ER marker (DsRed-HDEL), and a mitochondrial matrix-targeted blue fluorescent protein (mito-TagBFP). As shown in a representative series of time lapse images in *Figure 1C*, we observed ERMES foci at interfaces between the ER and mitochondria and subsequently at ER-associated mitochondrial constriction and eventual division events. Consistent with this observation, the yeast mitochondrial division machine, marked by Dnm1-mCherry, was observed adjacent to Mdm34-labeled ERMES foci at mitochondrial division sites (*Figure 1D*). Given that ERMES null mutations are epistatic to *DNM1* null mutations (*Youngman et al., 2004*), these observations suggest that ERMES functions early in ERMD at a step distinct from Dnm1 by bridging interactions between the ER and mitochondria.

The conserved Miro GTPase Gem1 associates with the ERMES complex, potentially as a regulatory subunit (*Kornmann et al., 2011*; *Stroud et al., 2011*). Consistent with this, a functional GFP-2xFLAG-Gem1 fusion protein expressed in *gem1Δ* cells localized to mitochondrial-associated foci labeled by the ERMES subunit Mdm34-mCherry (*Figure 1E*). Similar to ERMES, Gem1-labeled foci were spatially linked to mitochondrial division sites and segregated to only one tip following division (*Figure 1E*).

To gain insight into the function of Gem1 in ERMD, we characterized the behavior of ERMES and mitochondria in *gem1Δ* cells. As previously published, we observed that mitochondrial structure in *gem1Δ* cells is both aberrant and diverse, ranging from tubules to clustered spheres, with both having an increased mitochondrial diameter (*Figure 2A*) (*Frederick et al., 2004*). In addition, regardless of the mitochondrial structure type, we observed that ERMES foci in *gem1Δ* cells were associated with apparent mitochondrial constriction sites, defined as a narrowing and/or resolved separation of the mitochondrial matrix labeled by mito-DsRed (*Figure 2A*, arrow heads). A vast majority of ERMES-marked mitochondrial constriction sites in *gem1Δ* cells were stable, typically persisting for the duration of image capture (3–3.5 min) (*Figure 2B* and *Figure 2—figure supplement 1B,C*). This behavior was in contrast to wild-type cells, where ERMES-marked mitochondrial constriction sites resolved

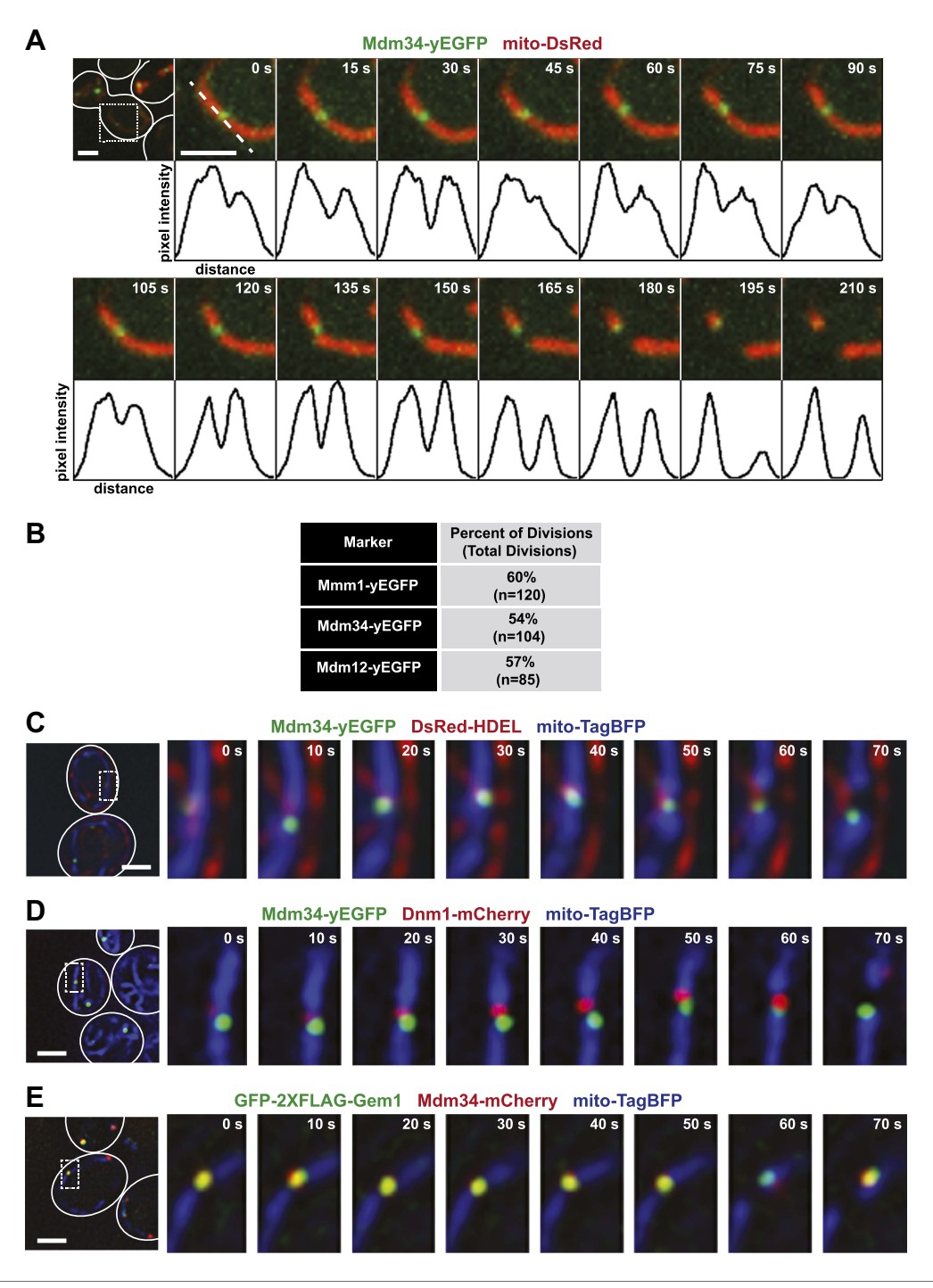

**Figure 1**. ERMES marks sites of ER-associated mitochondrial division. (**A**) A timelapse series of Mdm34-yEGFP and mito-DsRed. For clarity, the images represent a maximum projection of a 2-µm section of a cell that accurately represents a mitochondrial division observed in a single plane. The first frame represents a whole-cell projection containing an inset that is shown in the remaining frames. Below each frame of the timelapse series is a plot of pixel intensity vs distance (line-scan) of mito-DsRed fluorescence signal as indicated by the dashed line at 0 s. The range of the y-axis is 260 arbitratry units and the range of the x-axis is 151 pixels. (**B**) Quantification of ERMES-associated mitochondrial divisions in wild-type cells using data collected as described above. (**C**) ERMES foci are associated with ER tubules at mitochondrial division sites. Cells expressing Mdm34-yEGFP, DsRed-HDEL, and mito-TagBFP were imaged in a single plane every 10 s. The first frame represents a whole-cell projection

*Figure 1. Continued on next page*

*Figure 1. Continued*

containing an inset that is shown in the remaining frames. (**D**) ERMES foci are adjacent to Dnm1 at the mitochondrial division sites. Cells expressing Mdm34-yEGFP, Dnm1-mCherry, and mito-TagBFP were imaged in a single focal plane every 10 s (**E**) *gem1Δ* cells expressing GFP-2xFLAG-Gem1, Mdm34-mCherry, and mito-TagBFP were imaged in a single focal plane every 10 s. Expression of the mito-TagBFP construct in (**C–E**) was induced by growing the cells overnight to mid-log phase in synthetic media containing 2% galactose. Scale bars, 2 μm.

The following figure supplements are available for figure 1:

**Figure supplement 1**. ERMES subunits colocalize at mitochondrial division sites.

into two spatially separated tips within 15–60 s (*Figure 1A* and *Figure 2—figure supplement 1A,C*). These observations indicate that Gem1 function is required for the resolution of mitochondrial constriction sites associated with division and/or in the subsequent segregation of mitochondrial tips generated by division events.

We addressed what structural features are important for Gem1's role in the resolution of ERMES-associated mitochondrial constriction sites in cells. Under our conditions, expression of Gem1 mutants harboring abrogating mutations in either EF-hand I or II motifs or both rescued all mitochondrial phenotypes in *gem1Δ* cells to a similar extent as wild-type *GEM1* (*Figure 2—figure supplement 2*). This observation is consistent with the published work demonstrating that the EF-hand regions play a relatively minor role in the maintenance of mitochondrial morphology and distribution, with only EF-hand I acting to stabilize Gem1 expression levels (*Frederick et al., 2004*; *Koshiba et al., 2011*). In contrast, expression of Gem1S19N or Gem1S462N, whose mutations abrogate the GTPase activity of GTPase domains 1 and 2, respectively, failed to fully rescue mitochondrial morphological phenotypes in *gem1Δ* cells. The Gem1S19N mutation was the least functional, consistent with previously published observations (*Frederick et al., 2004*). Abrogating mutations in the first GTPase domain, such as the Gem1S19N mutation, have been shown to reduce the steady-state localization of Gem1 to ERMES foci (*Kornmann et al., 2011*). Thus, the severe phenotypes associated with Gem1S19N underscores the functional importance of Gem1's localization to ERMES foci. Significantly, we also observed stable ERMES-associated mitochondrial constrictions in *gem1Δ* cells expressing either Gem1S19N or Gem1S462N (*Figure 2—figure supplements 2D and 3A,B*). Thus, our structure function analysis of Gem1 indicates that similar features, specifically both Gem1 GTPase domains, are required for Gem1's roles in both the maintenance of mitochondrial morphology and in the resolution of ERMES-marked mitochondrial constriction sites. Thus, the resolution of ERMES-associated mitochondrial constriction sites into segregated mitochondrial tips is a central function of Gem1.

We addressed whether the stability of ERMES-marked mitochondrial constriction sites is a consequence of defective mitochondrial division in *gem1Δ* cells by examining the behavior of the division machine, labeled with Dnm1-mCherry, over time. We observed that stable mitochondrial constriction sites in *gem1Δ* cells (mito-TagBFP) were persistently associated with ER tubules (GFP-HDEL) (*Figure 2C*). In addition, Dnm1 puncta localized to these ER-associated mitochondrial constriction sites in a dynamic manner similar to that observed in wild-type cells, associating and dissociating on a 10-s time scale (*Figure 2C*) (*Frederick et al., 2004*). These observations indicate that neither ER engagement with nor Dnm1 recruitment to mitochondrial constriction sites is significantly altered in *gem1Δ* cells, suggesting that mitochondrial division per se is not defective. This conclusion is supported by the observation that in *gem1Δ* cells lacking Fzo1, an outer mitochondrial membrane dynamin-related protein required for mitochondrial fusion, mitochondria are fragmented, which is a morphology that requires mitochondrial division (*Frederick et al., 2004*). Thus, taken together, these observations indicate that the stable ERMES-associated mitochondrial constriction sites in *gem1Δ* cells is a consequence of defective resolution of mitochondrial tips, perhaps as a consequence of altered ER–mitochondria contacts.

To explore this idea, we analyzed wild-type and *gem1Δ* cells using three-dimensional electron tomography to determine whether mitochondrial constriction sites in *gem1Δ* cells represent division intermediates or fully resolved division events and to examine ER–mitochondria contact sites at higher resolution. Tomographic analysis revealed that, in contrast to tubular mitochondria in wild-type cells, mitochondria in *gem1Δ* cells are present in clusters containing fully resolved individual

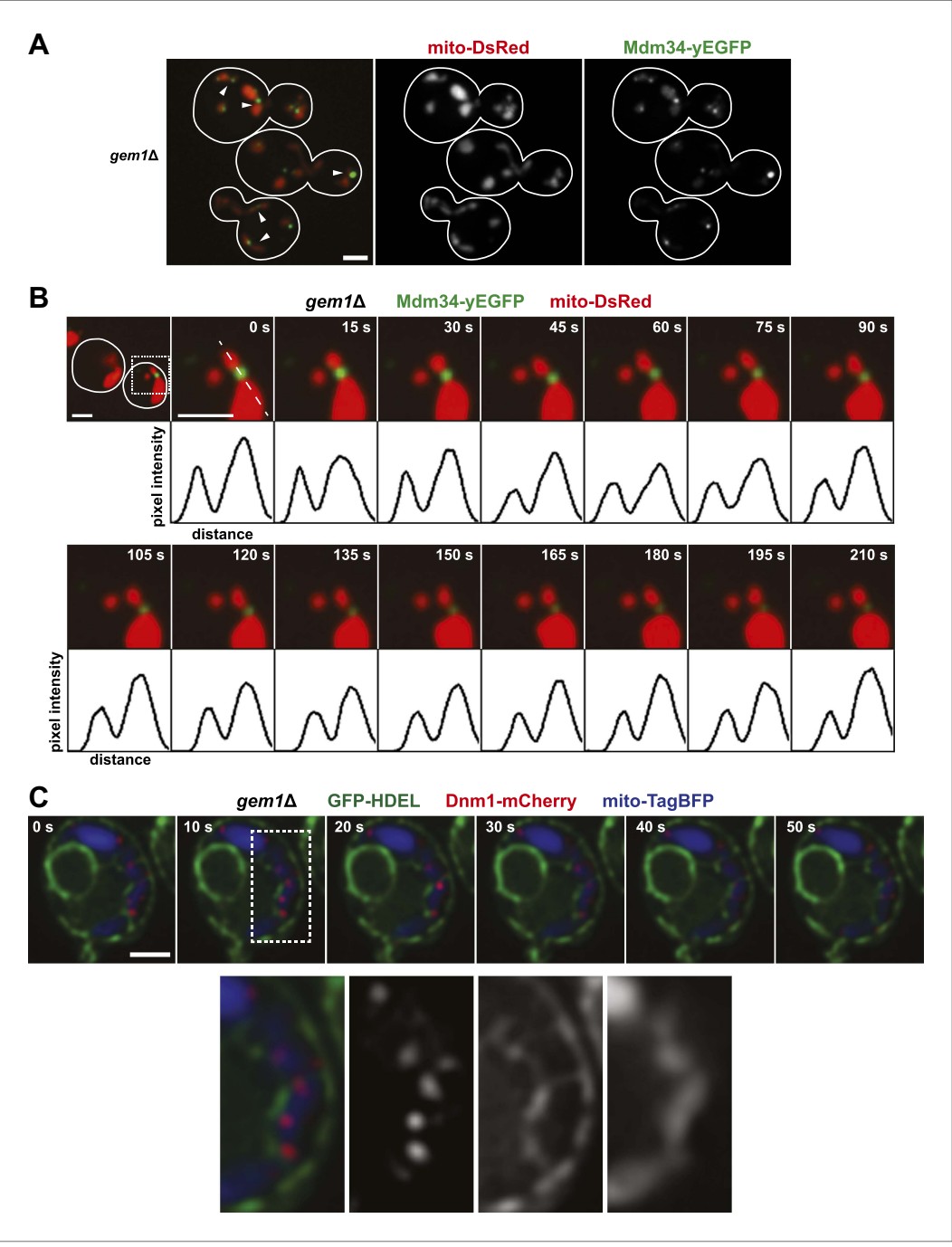

**Figure 2**. Gem1 is required for mitochondrial distribution during ER-associated mitochondrial division. (**A**) ERMES foci are found at mitochondrial constriction sites in *gem1*Δ cells. Depicted is a whole-cell projection of *gem1*Δ cells expressing Mdm34-yEGFP and mito-DsRed. Arrowheads indicate mitochondrial constriction sites, which are also associated with Mdm34-yEGFP labeled foci. (**B**) The constriction sites associated with ERMES in *gem1*Δ cells are stable. *gem1*Δ cells expressing Mdm34-yEGFP and mito-DsRed were imaged as in *Figure 1A*. The first frame represents a whole-cell projection containing an inset that is shown in the remaining frames. Below each frame of the timelapse series is a plot of pixel intensity vs distance (line-scan) of mito-DsRed fluorescence signal as indicated by the dashed line at 0 s. The range of the y-axis is 260 arbitrary units and the range of the x-axis is 151 pixels. (**C**) Dnm1 is targeted to mitochondria in *gem1*Δ cells. A single focal plan in a timelapse series of a *gem1*Δ cell expressing GFP-HDEL, mito-tagBFP, and Dnm1-mCherry is shown. ER tubules also stably associate with mitochondrial constriction sites. Scale bars, 2 µm.

*Figure 2. Continued on next page*

*Figure 2. Continued*

The following figure supplements are available for figure 2:

**Figure supplement 1**. ERMES-marked mitochondrial division is attenuated in *gem1Δ* cells.

**Figure supplement 2**. The first and second GTPase domains of Gem1 are required for its role in maintaining mitochondrial morphology.

**Figure supplement 3**. GTP hydrolysis by the second GTPase domain of Gem1 is required for mitochondrial distribution at ERMES-linked constrictions.

organelles (*Figure 3A–D* and *Videos 1–4*). Importantly, fully separate and adjacent mitochondria interacted with the same ER segment (*Figure 3B,C*, $B_2$ and $C_1$, ER–mitochondrial contact sites in red), suggesting that after division, resolved mitochondria remain tethered via this shared ER segment.

Quantitative analysis of an equivalent mitochondrial surface area in wild-type and *gem1Δ* cells (approximately 5 $\mu m^2$) revealed that *gem1Δ* cells harbor a significantly greater number of ER–mitochondria contacts than wild-type cells, which are also more clustered as compared to wild-type contacts (*Figure 3—figure supplement 1*). The surface area of individual ER–mitochondria contacts was, on average, smaller in *gem1Δ* cells as compared to wild-type cells (*Figure 3E* and *Figure 3—figure supplement 1C*). However, the total ER–mitochondria contact surface was approximately three times greater than that observed for wild-type mitochondria (*Figure 3F*). These findings are consistent with fluorescence data demonstrating that, in comparison to wild-type cells, ER tubules in *gem1Δ* cells were observed associated in a relatively stable manner with mitochondrial constriction sites (*Figure 2C*). In addition, our EM analysis suggests that the reported larger ERMES foci observed by fluorescence imaging in *gem1Δ* cells may represent multiple unresolved ER–mitochondria contacts (*Kornmann et al., 2011*) (*Figure 3—figure supplement 1D*). The average diameter of *gem1Δ* mitochondria, measured along their short axis, was also greater than that of wild-type cells (*Figure 3—figure supplement 1E*), consistent with our fluorescence imaging analysis (*Figure 2*). However, the average diameter of ER-associated mitochondrial constriction sites in wild-type and *gem1Δ* cells were similar (*Figure 3—figure supplement 1F*), consistent with our light imaging data demonstrating that Dnm1 is still efficiently targeted to mitochondrial constriction sites in *gem1Δ* cells (*Figure 2C*). Together, these data suggest that Gem1 functions downstream of ER-associated mitochondrial constriction and division as a negative regulator of ER–mitochondria contacts, which is required for the efficient resolution and spatial distribution of mitochondria following division.

To further investigate the functional significance of these observations, we explored the idea that ERMD serves to position division sites adjacent to actively segregating mitochondrial nucleoids, which is based on the observation that ERMES is spatially linked to actively replicating mitochondrial nucleoids (*Hobbs et al., 2001*; *Meeusen and Nunnari, 2003*). To visualize nucleoids, we created a functional FP fusion to the nucleoid component, Yme2 (*Figure 4* and *Figure 4—figure supplements 1 and 2*), an integral mitochondrial inner membrane protein with homology to exonucleases whose deletion alters nucleoid structure and copy number (*Hanekamp and Thorsness, 1996*, *1999*; *Park et al., 2006*). Consistent with the previously published work, we observed that over time a subset of Yme2-GFP–labeled nucleoids persistently localize adjacent to ERMES foci, marked by Mdm34-tdTomato (*Figure 4A*). In addition, ERMES-linked nucleoids were associated with mitochondrial division events (*Figure 4A*). To characterize the behavior of yeast nucleoids at division sites, we simultaneously imaged mitochondria (mito-dsRed) and nucleoids (Yme2-GFP). Quantitative analysis indicated that mitochondrial nucleoids are associated with over 80% of mitochondrial division events (*Figure 4B,C*). Temporal analysis of Yme2-GFP–labeled nucleoids positioned at future sites of mitochondrial division indicated that they exhibit short-ranged oscillatory movements and often rapidly segregate and recoalesce prior to mitochondrial division (*Figure 4B* and *Figure 4—figure supplement 3*). Following mitochondrial division, a majority of Yme2-GFP foci were observed at the ends of both newly generated mitochondrial tips (*Figure 4B,C* and *Figure 4—figure supplement 3*). However, we also observed division events that resulted in a nucleoid in only one of the newly generated mitochondrial tips, indicating that division and nucleoid segregation are not obligatorily linked (*Figure 4C* and *Figure 4—figure supplement 3*). Consistent with a role of ERMES in nucleoid positioning, published

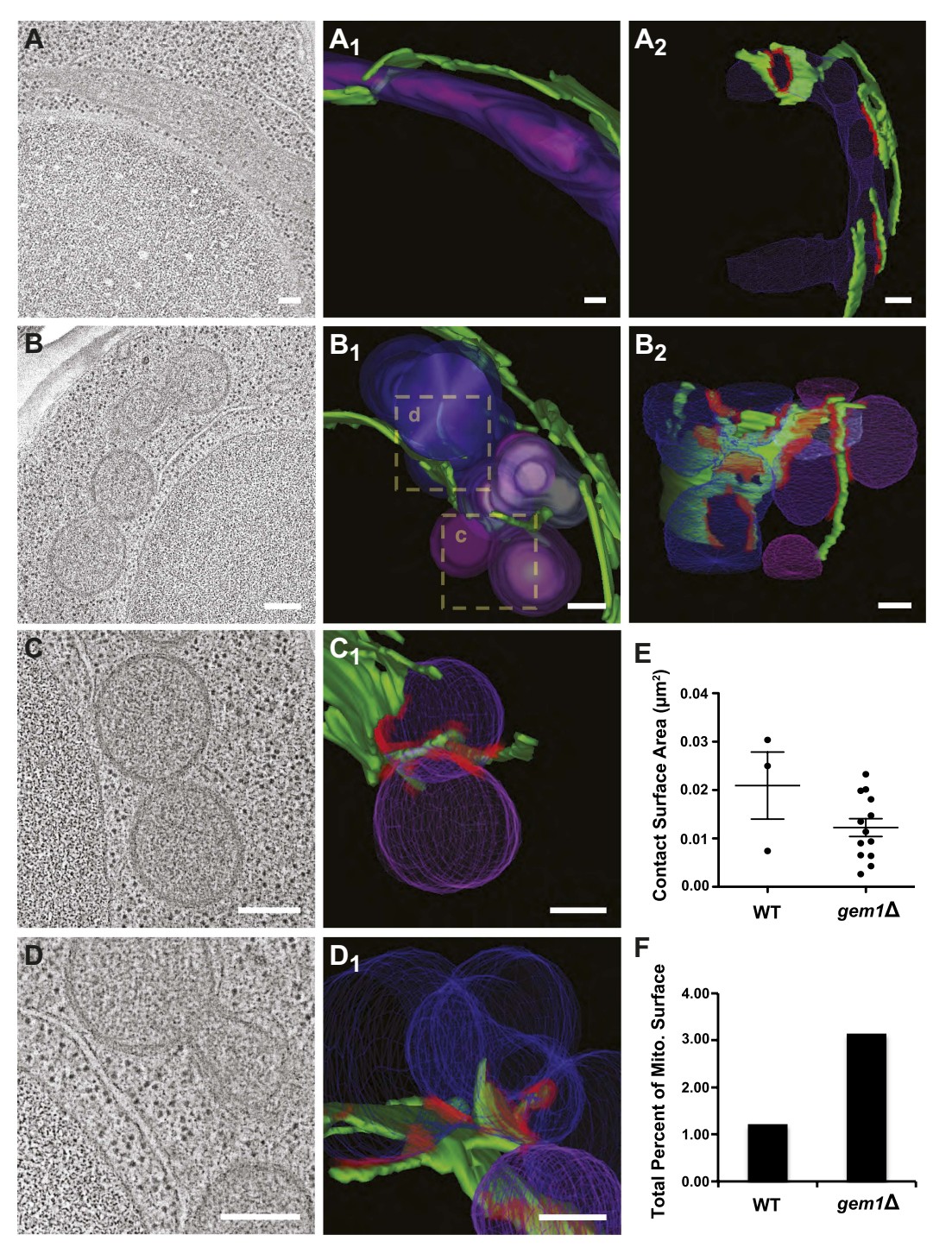

**Figure 3**. Gem1 regulates ER–mitochondria contacts. (**A**) A tomograph and (**A₁** and **A₂**) corresponding three-dimensional tomogram of a mitochondrion (purple) and the ER (green) that closely apposes it (contact sites in red, defined as <30 nm distance and ribosome excluded) in a wild-type yeast cell. (**B**) Tomograph and (**B₁** and **B₂**) three-dimensional tomogram of mitochondria (pink and purple) and ER (green, with contact shown in red) for a *gem1Δ* cell. Mitochondria are shown in multiple hues of purple to indicate mitochondria that are discontinuous within the reconstructed volume. Magnified tomographs and three-dimensional tomograms of the purple mitochondria (**C** and **C₁**) and blue mitochondria (**D** and **D₁**) shown in the boxed regions of B₁ are shown. (**E**) Tomograms in (A, wild type) and (B, *gem1Δ*) were used to calculate the area of mitochondrial surfaces closely apposed to the ER (<30 nm, excludes ribosomes). The total mitochondrial surface area analyzed is similar for wild-type and *gem1Δ* cells (4.85 and 5.09 µm², respectively). Cells lacking Gem1 possess more clustered and smaller ER–mitochondria contacts. (**F**) The total surface area was calculated for the mitochondria that

*Figure 3. Continued on next page*

*Figure 3. Continued*

were modeled and shown in (A, wild-type) and (B, *gem1*Δ), and the percent of this surface area that was in contact with the ER membrane was calculated for each. Mean diameter of the mitochondria shown in (A, wild-type) and (B, *gem1*Δ). Scale bars, 200 nm.

The following figure supplements are available for figure 3:

**Figure supplement 1**. Quantitative comparison of ER–mitochondria contacts in wild-type and *gem1*Δ cells.

data indicate that deletion of core ERMES components disrupts nucleoid structure (***Burgess et al., 1994***; ***Youngman et al., 2004***). In contrast, we observed that nucleoids, as labeled by Yme2-GFP, maintained their discrete focal localization in *gem1*Δ cells, similar to wild-type cells (***Figure 4D***). In addition, similar to ERMES distribution, Yme2-GFP foci were observed associated with stable mitochondrial constrictions in *gem1*Δ cells (***Figure 4D***), consistent with the idea that the spatial resolution of ER–mitochondria contacts following division is fundamentally important for nucleoid distribution. Thus, our findings suggest that ERMD functions to position division sites in proximity to nucleoids to increase the probability of their distribution upon mitochondrial division.

## Discussion

Here, we show that ERMES and the Miro GTPase Gem1 function in the process of ERMD, which serves to couple the segregation of mitochondria and mtDNA in cells. Specifically, our data support a model in which ERMES creates ER–mitochondria contacts along the mitochondrial network that serve to link actively replicating nucleoids to mitochondrial division and the subsequent Gem1-dependent spatial resolution of newly generated mitochondrial tips. Such a mechanism is likely required for efficient mtDNA distribution throughout the cell in addition to nucleoid segregation *per se*, as long-range movement of nucleoids within the organelle is limited (***Nunnari et al., 1997***; ***Okamoto et al., 1998***). In mammalian cells, nucleoids are similarly localized at mitochondrial division sites and mitochondrial tips, suggesting that ERMD also plays this fundamental role in humans (***Garrido et al., 2003***; ***Iborra et al., 2004***). In this context, although the Miro GTPase family is highly conserved in eukaryotes, Miro orthologs are not found in organisms that lack mtDNA, such as *Giardia intestinalis* and *Trichomonas vaginalis*, and possess mitochondrial-related mitosomes or hydrogenosomes, respectively (***Vlahou et al., 2011***). This correlation is consistent with a fundamental role of Miro in mtDNA segregation. Also consistent with this view, *gem1*Δ cells lose mitochondrial DNA at a significantly higher frequency than wild-type cells (***Frederick et al., 2004***).

The mechanism of mitochondrial division site placement is apparently divergent from that utilized by ancestral bacteria, where division sites are determined in part by a nucleoid occlusion mechanism, which prevents cell division in the vicinity of the bacterial chromosome (***Wu and Errington, 2012***). However, we currently lack an understanding of what drives mitochondrial nucleoid segregation and of the mechanism underlying the spatial coupling of ER–mitochondria contact sites and nucleoids. Thus, alternative mitochondrial-specific mechanisms may exist to coordinate the timing of nucleoid replication and segregation with mitochondrial division.

In ERMD, mitochondrial constrictions observed at the sites of ER–mitochondria contact, where the ER likely wraps around a mitochondrial tubule, are independent of the mitochondrial division dynamin (***Friedman et al., 2011***). Our data are consistent

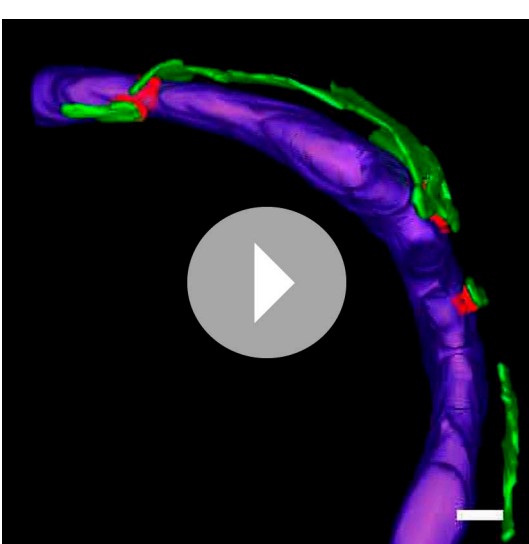

**Video 1**. Original tomographs and rotating three-dimensional models of the ER and mitochondria in a wild-type cell. ER (green) and a mitochondrion (purple) in a tomogram derived from three serial sections of a wild-type cell. Shown in red are regions of contact between the two organelles.

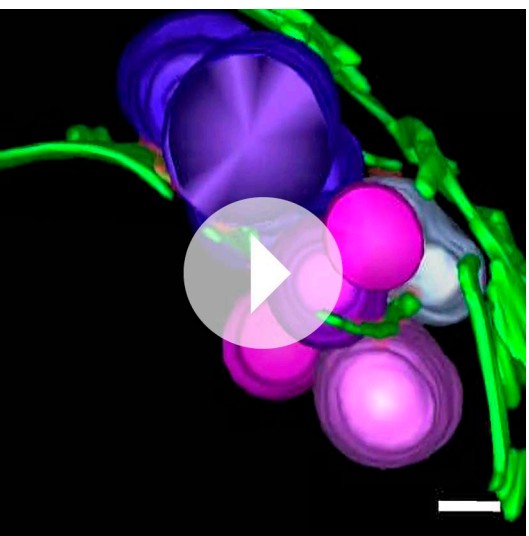

**Video 2**. Original tomographs and rotating three-dimensional models of ER and mitochondria in a *gem1Δ* cell. ER (green) and a mitochondrion (purple) in a tomogram derived from three serial sections of a *gem1Δ* cell. Shown in red are regions of contact between the two organelles.

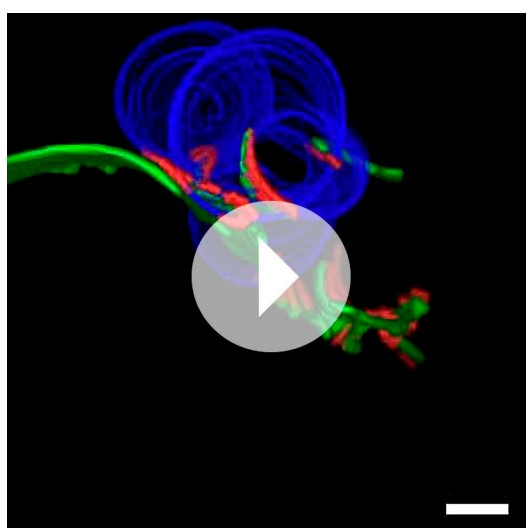

**Video 3**. A rotating three-dimensional model of constricted, but continuous mitochondria in a *gem1Δ* cell. Rotating three-dimensional models of mitochondria (purple) that are globular and yet continuous with each other relative to the ER (green) and regions of contact between them (red) in the *gem1Δ* cell. Images correspond to the mitochondria shown in *Figure 1B,C*.

with the model in which ERMES functions early in ERMD to mediate the biogenesis of this specialized region of ER–mitochondria contact. Thus, it is tempting to speculate that the ERMES complex functions to generate the ER tubules associated with constriction and/or to actively produce mitochondrial constriction at the sites of contact. ERMES has also been implicated as a bridge between mitochondria and the actomyosin network and thus may function at division sites to coordinate the recruitment of cytoskeletal and motor proteins, which could generate force required for mitochondrial constriction and/or distribution following division (*Boldogh et al., 1998*, *2003*). Indeed, a recent study suggests that the actin cytoskeleton may be involved in mitochondrial division at ER–mitochondria contacts in mammalian cells (*Korobova et al., 2013*).

In contrast to ERMES, our data indicate that Gem1 functions relatively late in ERMD to promote the physical separation of mitochondrial tips generated by membrane scission. Specifically, our data point to a role for Gem1 post-scission in the negative regulation of ER–mitochondria contacts to facilitate the resolution of mitochondrial tips. Evidence from higher eukaryotes suggests that the Gem1 ortholog, Miro1/2, functions in mitochondrial distribution by connecting mitochondria directly to a kinesin-1 adaptor protein Milton/TRAK to enable the microtubule based transport of mitochondria (*Guo et al., 2005*; *Fransson et al., 2006*; *Glater et al., 2006*; *Wang and Schwarz, 2009*; *Misko et al., 2010*; *Wang et al., 2011*). Analogously, Gem1 in yeast may also function to negatively regulate ER–mitochondria contacts by facilitating the recruitment of motility factors to mitochondrial tips following division. However, while the Miro GTPase family is remarkably conserved (*Vlahou et al., 2011*), the mechanisms of mitochondrial transport are divergent in eukaryotes. In many fungi, including budding yeast, mitochondrial distribution is actin dependent (*Hermann et al., 1997*; *Simon et al., 1997*; *Boldogh et al., 1998*, *2003*), and in *Dictyostelium*, Miro is not required for microtubule-dependent mitochondrial transport (*Vlahou et al., 2011*). Thus, alternatively, Gem1 may function in ERMD more directly to regulate the physical link between mitochondria and the ER via ERMES. Consistent with this idea, Gem1 associates with ERMES foci, and, in the absence of Gem1, it has been reported that ERMES forms fewer and larger foci per cell

(*Kornmann et al., 2011*). This model of Gem1 function is also supported by the observation that mutations in the first GTPase domain of Gem1 abolish both Gem1's function in ERMD and its association with ERMES (*Frederick et al., 2004*; *Kornmann et al., 2011*; *Koshiba et al., 2011*). In addition, in mammalian cells, Miro1 is found at ER–mitochondria contacts (*Kornmann et al., 2011*). In this

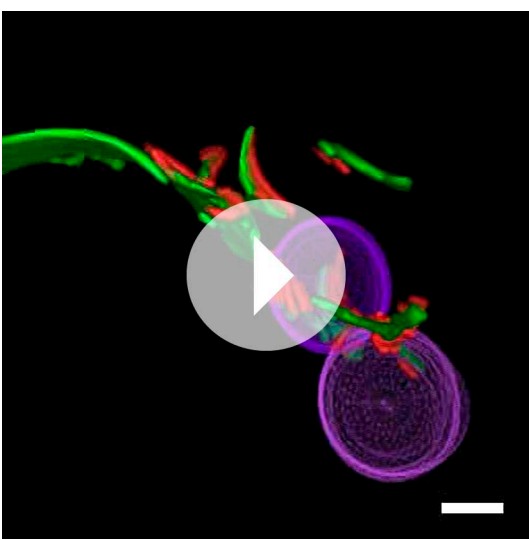

**Video 4**. A rotating three-dimensional model of 'tethered' mitochondria in a *gem1Δ* cell. Globular and 'tethered' mitochondria in a *gem1Δ* cell are associated with the ER (green) and regions of contact between them (red) are noted. Images correspond to the mitochondria shown in *Figure 1B,D*.

context, the defect in mitochondrial segregation observed in *gem1Δ* cells at ER-associated mitochondrial constriction sites could result from the ER physically hindering mitochondrial motility. In this case, it is also possible that the mitochondrial motility defects in higher eukaryotes caused by Miro1/2 dysfunction are a secondary consequence of enhanced ER–mitochondria contacts.

## Materials and methods

### Yeast strains and plasmids

All strains were constructed in either the W303 (*ade2–1*; *leu2–3*; *his3–11, 15*; *trp1–1*; *ura3–1*; *can1–100*) or BY4741 (*his3Δ1*; *leu2Δ0*; *met15Δ0*; *ura3Δ0*) genetic background, both of which have been described previously (*Rothstein, 1983*; *Brachmann et al., 1998*). ERMES components and Yme2 were tagged at their C-termini using PCR-based homologous recombination (*Janke et al., 2004*; *Sheff and Thorn, 2004*). The red fluorescent protein TdTomato (*Shaner et al., 2004*) was cloned into pFA6 plasmids using standard cloning techniques. Haploid cells were transformed with PCR product using the lithium acetate method and plated on synthetic media or YPD + geneticin (300 µg/ml) to select for the homologous recombination event. Correct integration of the cassette was confirmed by colony PCR, and western blotting of whole-cell protein extracts was used to verify correct protein expression. To generate *gem1Δ MDM34-yEGFP* or *gem1Δ MDM34-mCherry* strains, W303 *MDM34-yEGFP::HIS5* or W303 *MDM34-mCherry* was mated to *gem1Δ::HIS3* cells. The heterozygous diploids were sporulated, and the resulting tetrads were dissected. Two-to-two cosegregation of the histidine prototrophy was used to verify the *gem1Δ MDM34-yEGFP* genotype, and cosegregation of histidine prototrophy and kanamycin resistance was used to verify the *gem1Δ MDM34-mCherry* genotype. The *yme2Δ gep4Δ* and *YME2-GFP gep4Δ* strains were generated by mating haploid *gep4Δ* and *yme2Δ* or *YME2-GFP* strains, followed by tetrad dissection and identification of clones by cosegregating genetic markers.

The plasmids pVT100U-DsRed (mito-DsRed), pHS20-mCherry (Dnm1-mCherry), pRS315-GFP-2xFLAG-GEM1 (GFP-2XFLAG-Gem1), and YIplac204/TKC-DsRed-HDEL (DsRed-HDEL) YIplac204/TKC-GFP-HDEL (GFP-HDEL) were described previously (*Westermann and Neupert, 2000*; *Rossanese et al., 2001*; *Meeusen and Nunnari, 2003*; *Lackner et al., 2009*; *Kornmann et al., 2011*). The plasmid pYES-TagBFP (mito-TagBFP) was obtained by replacing BFP in pYES-BFP (*Westermann and Neupert, 2000*) with TagBFP using standard cloning techniques.

Plasmids harboring wild-type and mutant alleles of *GEM1* were generated by amplifying the *GEM1* locus from yeast genomic DNA with flanking regions 307-bp upstream and 644-bp downstream by PCR followed by insertion into pRS315 with standard restriction cloning techniques. Point mutations were generated using Quick-Change Mutagenesis, and all mutations were verified by DNA sequencing. Plasmids were transformed into *gem1Δ MDM34-yEGFP* pVT100U-DsRed grown to mid-log phase in synthetic ethanol glycerol media at 23°C.

### Light microscopy and image processing

For all fluorescence microscopy described in the next two paragraphs, yeast cells were grown to mid-log phase in the appropriate synthetic media and were imaged live. Yeast cells grown to mid-log phase in the appropriate synthetic media were briefly sonicated, concentrated by centrifugation, and mounted on slides with a 4% agarose bed in synthetic dextrose growth medium.

To evaluate the relationship of ERMES to mitochondrial division, wild-type or *gem1Δ* cells expressing Mdm12, Mdm34, or Mmm1-yEGFP and mito-DsRed were imaged using the spinning disc module of

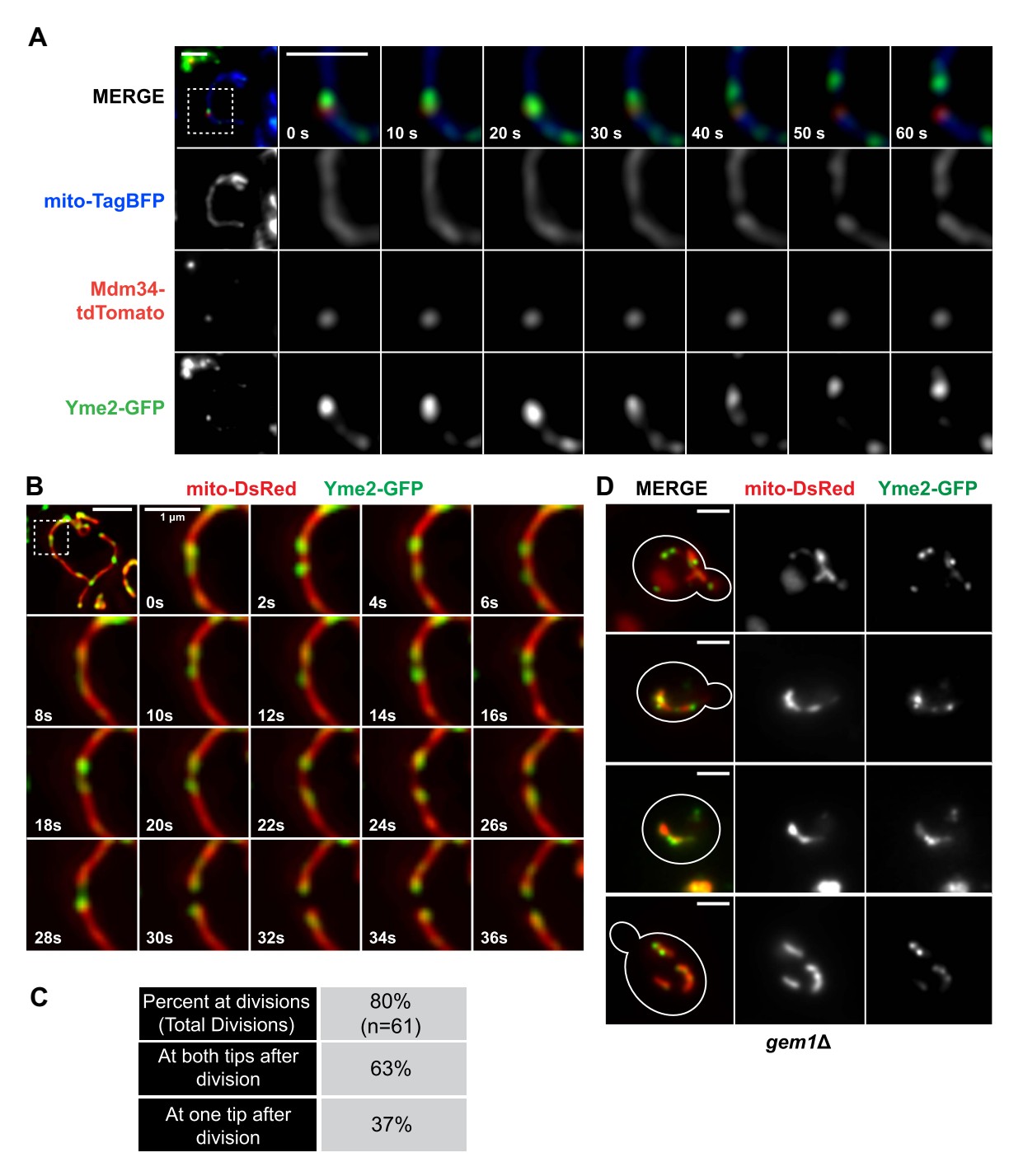

**Figure 4**. Nucleoid segregation is linked to mitochondrial division. (**A**) Yme2-GFP coaligns with ERMES foci, marked by Mdm34-TdTomato, at mitochondrial division events. (**B**) Nucleoid behavior, marked by Yme2-yEGFP, at mitochondrial (mito-DsRed) division sites. (**C**) Quantification of the relationship of nucleoids to division sites. (**D**) Representative images of *gem1Δ* cells expressing Yme2-GFP and mito-DsRed. Scale bars, 2 μm, except in the inset of (**B**), which is 1 μm.

The following figure supplements are available for figure 4:

**Figure supplement 1**. Yme2-GFP is functional.

**Figure supplement 2**. Yme2-GFP localizes to nucleoids.

**Figure supplement 3**. Additional examples of Yme2-GFP nucleoid behavior at a mitochondrial division sites in wild-type cells.

a Marianas SDC Real Time 3D Confocal-TIRF microscope (Intelligent Imaging Innovations Denver, CO, 3i) fit with a Yokogawa spinning disk head, a 100× 1.46 NA objective, and EMCCD camera. Z-stacks were taken at 0.4-μm increments over approximately 6 μm of the cell every 15 s for 3.5 min. Mitochondrial divisions were identified independently of ERMES foci by hiding the GFP channel and retrospectively analyzed for their association with ERMES. An ERMES focus was considered linked to mitochondrial division if its center was within 300 nm of the division site, which is 1.5 times the average radius of an ERMES focus (205 ± 45 nm, n = 68). To calculate the ERMES to mitochondrial surface area ratio, Z-series of cells expressing Mdm34-yEGFP and mitoDsRed were collected at 0.4-μm increments. Separate channels for each marker were manually threshold in ImageJ, and the 'Analyze Particles' function was used to measure areas for each in all focal planes collected. Ratios were calculated for each plane and were averaged. This ratio (4.5%) was used to calculate the percentage of ERMES foci associated with mitochondrial divisions due to random chance (approximately 10%) based on the requirement that mitochondrial division occurred less than 300 nm from the center of an ERMES focus.

For three-color fluorescence imaging in *Figures 1C-E and 2C*, cells were imaged on DeltaVision-Real Time microscope (IX70 DeltaVision; Olympus) using a 60× 1.4 NA objective lens (Olympus) and a 100 W mercury lamp (Applied Precision). Light microscopy images were collected using an integrated cooled charge-coupled device (CCD)–based camera (CoolSNAP HQ; Photometrics) equipped with a Sony Interline Chip. Datasets were processed using softWoRx's (Applied Precision) iterative, constrained three-dimensional deconvolution method to remove out-of-focus light.

To analyze *GEM1* alleles by fluorescence microscopy, *gem1Δ* MDM34-yEGFP pVT100U-DsRed pRS315-(empty/*GEM1*/*gem1* alleles) cells were grown to early log phase, concentrated by centrifugation and mounted on slides with 4% agarose beds and imaged on the Marianas SDC Real Time 3D Confocal-TIRF microscope as described above.

For all imaging of Yme2-GFP, cells were grown to mid-log in synthetic dextrose medium supplemented with casamino acids. Cells were immobilized in glass bottomed culture dishes (Bioptechs, Inc) with Concanavalin A (1 mg/ml) and overlaid with 2 ml of fresh medium. Cells were imaged on an OMX microscope equipped with a 100× 1.4 NA objective lens (*Carlton et al., 2010*). Z-stacks were acquired over 7 μm in 0.2-μm increments. For higher temporal resolution (2 s per frame), only 3-μm-thick Z-stacks were collected. Images for each fluorophore were acquired simultaneously with independent EMCCD cameras (iXON; Andor), and images were aligned post-capture using alignment parameters generated from images of 0.1-μm fluorescent microspheres (TetraSpeck; Invitrogen). Images were processed with a denoising algorithm (*Boulanger et al., 2010*) and iterative, constrained three-dimensional deconvolution using the Priism software suite (http://msg.ucsf.edu/IVE/). Yme2-associated division events were scored as described above for Mdm34-yEGFP. For DAPI staining, cells were grown for 30 min in synthetic dextrose medium supplemented with casamino acids and 1 μg/ml DAPI and subjected to imaging.

## Electron microscopy and tomography

Haploid *Saccharomyces cerevisiae* cells were grown to log phase, harvested, and high-pressure frozen in a Balzers HPM 010 as previously described (*Nickerson et al., 2010*). Automated Freeze Substitution was performed on a Leica AFS with 0.1% uranyl acetate and 0.25% glutaraldehyde in anhydrous acetone, embedded in Lowicryl HM20 (Polysciences, Warrington, PA), and polymerized at −60°C (*Giddings, 2003*). A Leica Ultra-Microtome was used to cut 300-nm serial semi-thick sections; sections were stabilized using a formvar sandwich (*West et al. 2011*) and labeled with fiduciary 15-nm colloidal gold (British Biocell International); dual-axis tilt series were collected of the samples from ±60° with 1° increments at 300 kV using SerialEM (*Mastronarde et al., 1997*) at 300 kV using a Tecnai 30 FEG (FEI-Company, Eindhoven, the Netherlands). Tilt series were recorded at a magnification of 23,000 times using SerialEM with a 4 × 4K CCD camera (Gatan, Inc., Abingdon, United Kingdom) as described (*West et al. 2011*). Individual tomograms were reconstructed using the IMOD package (*Kremer et al, 1996*), and its newest viewer 3DMOD 4.0.11 and serial tomograms were merged together in X-, Y-, and Z-direction to obtain a large continuous volume. The 3DMOD modeling software was used for the assignment of the outer leaflet of organelle membrane contours, and IMODINFO was used to obtain surface area and volume data of contour models. Images were further enhanced and manipulated in Adobe Photoshop 7. We sorted, analyzed, and graphed the data using Microsoft Excel for Mac 2008 and Prism 5 for Mac OS X. Movies were made in 3DMOD, assembled in QuickTime Pro 7.5, and movie size was reduced to less than 10 MB by saving movies as HD 720p

in QuickTime. Mitochondrial surface area was scored as in contact with the ER membrane (denoted as red objects in the three-dimensional models) if it was within 30 nm, and ribosomes were excluded between the two membranes.

## Acknowledgements

We thank Benoit Kornmann for sharing unpublished data and for helpful discussions and comments on the manuscript. We also thank members of the Nunnari Laboratory for providing comments and discussions and Dr Michael Paddy in the Molecular and Cellular Biology Imaging Facility at University of California, Davis, Drs John Sedat and Jennifer Fung at the University of California, San Francisco for technical assistance with fluorescence microscopy.

## Additional information

### Competing interests

JN, Reviewing editor, *eLife*. The other authors declare that no competing interests exist.

### Funding

| Funder | Grant reference number | Author |
| --- | --- | --- |
| National Institutes of Health | 5T32GM007377-34 | Andrew Murley |
| Howard Hughes Medical Institute | | Peter Walter |
| Human Frontier Science Program | | Christof Osman |
| National Institutes of Health | R01GM097432 | Laura L Lackner, Jodi Nunnari |
| National Institutes of Health | RO1GM083977 | Matthew West, Gia K Voeltz |

The funders had no role in study design, data collection and interpretation, or the decision to submit the work for publication.

### Author contributions

AM, GKV, PW, Conception and design, Acquisition of data, Analysis and interpretation of data, Drafting or revising the article; LLL, CO, Acquisition of data, Analysis and interpretation of data, Drafting or revising the article; MW, Acquisition of data, Analysis and interpretation of data; JN, Conception and design, Drafting or revising the article

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
