## [Decision Letter]

Thank you for choosing to send your work entitled “ER-associated mitochondrial division links the distribution of mitochondria and mitochondrial DNA” for consideration at *eLife*. Your article was found to be rigorous and very interesting after evaluation by a Senior editor and 3 reviewers, one of whom is a member of our Board of Reviewing Editors.

The Reviewing editor, Richard Youle, and the other reviewers discussed their comments before we reached this decision, and the Reviewing editor has assembled the following comments to help you prepare a revised submission.

1) The role of Gem1 for the inheritance of mt-DNA remains less developed, especially as division and nucleoid segregation appear not to be linked obligatorily. Does a deletion of GEM1 affect the distribution of YME2-GFP?

2) It has been reported that the GTPase domain and the first calcium binding domain of Gem1 might be differentially involved in ERMES foci localization and lipid exchange (23). It would be important to determine if either of these Gem1 mutations would rescue the unresolved ERMES complex phenotypes of *Δgem1* deletion strains.

3) In panels such as Figure 1, the authors describe ERMES at sites of ERMD. However, it is not clear that these mitochondria have not already divided by time 0, and are merely tethered to one another, and what we are seeing is the break of the tether. This might be more obvious if the individual channels for the 0 time point could be reviewed. This seems to be shown better in Figure 1, but the putative division events are hard to see. Perhaps the authors could do an enlargement for panel C as they did in panel A?

4) As above, in Figure 2 one has trouble establishing whether these sites are mitochondrial constriction sites vs separate mitochondria with an ER tubule between them. Perhaps a series of enlargements below would help to clarify this.

5) It was shown previously that ERMES foci are larger in size and fewer in number in *Δgem1* mutants (23). However, the authors find that *Δgem1* mutants have more ER-mitochondrial contact sites but the surface area of individual contacts is smaller (Figure 5C). Are some of the greater number of ER-mitochondrial contact sites in *Δgem1* mutants lacking in ERMES complexes? And are the large contact areas in *Δgem1* mutants lacking ERMES proteins from most of the interface? Or is there a discrepancy between this aspect of Kornmann’s paper and the current work? Some comment on this in the manuscript seems warranted.

6) During cell division in bacteria, constriction sites are often formed at a distal position relative to nucleoids. This is thought to prevent shedding of the genome and facilitate DNA segregation to daughter cells. As the authors argue differently for the inheritance of mt-DNA, some comments on the differences might be helpful.

---

## [Author Response]

*1) The role of Gem1 for the inheritance of mt-DNA remains less developed, especially as division and nucleoid segregation appear not to be linked obligatorily. Does a deletion of GEM1 affect the distribution of YME2-GFP*?

We analyzed the distribution of Yme2-GFP foci in *Δgem1* cells and we have added representative images to Figure 4. We observed that like ERMES, Yme2-GFP is predominantly localized to mitochondrial constrictions, consistent with our proposed role of ERMD in nucleoid distribution.

*2) It has been reported that the GTPase domain and the first calcium binding domain of Gem1 might be differentially involved in ERMES foci localization and lipid exchange (23). It would be important to determine if either of these Gem1 mutations would rescue the unresolved ERMES complex phenotypes of* Δgem1 *deletion strains*.

We addressed what structural features are important for Gem1’s role in the resolution of ERMES-associated mitochondrial constriction sites in cells. These data are presented as Figure 2—figure supplement 2. Our structure function analysis of Gem1 indicates that similar features, specifically both Gem1 GTPase domains, are required for Gem1’s roles in both the maintenance of mitochondrial morphology and in the resolution of ERMES-marked mitochondrial constriction sites into segregated mitochondrial tips. Thus, the resolution of ERMES-associated mitochondrial constriction is a central function of Gem1.

*3) In panels such as Figure 1, the authors describe ERMES at sites of ERMD. However, it is not clear that these mitochondria have not already divided by time 0, and are merely tethered to one another, and what we are seeing is the break of the tether. This might be more obvious if the individual channels for the 0 time point could be reviewed. This seems to be shown better in Figure 1, but the putative division events are hard to see. Perhaps the authors could do an enlargement for panel C as they did in panel A*?

*4) As above, in Figure 2 one has trouble establishing whether these sites are mitochondrial constriction sites versus separate mitochondria with an ER tubule between them. Perhaps a series of enlargements below would help to clarify this*.

To address points 3 and 4, we constructed plots of pixel intensity vs distance (line-scan) of the mito-DsRed fluorescence signal of ERMES-associated mitochondrial division and constriction sites in wild-type and *Δgem1* cells in revised Figures 1 and 2, respectively. These line scans have been placed below each of the corresponding image frames and clearly show that: A) For the wild-type event, mitochondria have not already divided by time 0; B) The “constrictions” we observe in *Δgem1* cells are stable, i.e., do not segregate, and, when compared to wild type division sites, are consistent with mitochondria that have already divided. However, fluorescence images lack sufficient resolution to make this conclusion. These line scans were performed for all of the examples shown in Figure 2—figure supplement 1 and validated the quantification shown in part C of this figure supplement, but were not included.

*5) It was shown previously that ERMES foci are larger in size and fewer in number in* Δgem1 *mutants (23). However, the authors find that* Δgem1 *mutants have more ER-mitochondrial contact sites but the surface area of individual contacts is smaller (Figure 5). Are some of the greater number of ER-mitochondrial contact sites in* Δgem1 *mutants lacking in ERMES complexes? And are the large contact areas in* Δgem1 *mutants lacking ERMES proteins from most of the interface? Or is there a discrepancy between this aspect of Kornmann’s paper and the current work? Some comment on this in the manuscript seems warranted*.

The differences in ERMES foci reported by Kornmann et al. were not large. We did not extensively analyze the difference in ERMES foci in wild type and *Δgem1* cells, but the trend we observed was consistent with that previously reported. Probing the differences in ERMES in *Δgem1* cells will require the development of more dynamic and more quantitative assays that are beyond the scope of this manuscript.

The increased number of smaller contacts may be below detection by fluorescence microscopy, but we cannot exclude the possibility that some ER-mitochondria contacts are ERMES-independent. Indeed, we anticipate that there are many different molecular types of ER-mitochondria contacts in cells and these may be modulated in response to loss of Gem1. In addition, we observed that the distance between any pair of ER-mitochondria contacts is decreased in *Δgem1* cells. This distance approaches diffraction-limited resolution of fluorescence microscopy, raising the possibility that the brighter ERMES foci observed by fluorescence microscopy in *Δgem1* cells could represent multiple unresolved ER-mitochondria contacts.

*6) During cell division in bacteria, constriction sites are often formed at a distal position relative to nucleoids. This is thought to prevent shedding of the genome and facilitate DNA segregation to daughter cells. As the authors argue differently for the inheritance of mt-DNA, some comments on the differences might be helpful*.

We have included a short paragraph in the Discussion about the potential divergence of division site placement mechanisms in mitochondria and bacteria.